# Learning Goal-Oriented Non-Prehensile Pushing in Cluttered Scenes

Nils Dengler          David Großklaus          Maren Bennewitz

*Abstract*— Pushing objects through cluttered scenes is a challenging task, especially when the objects to be pushed have initially unknown dynamics and touching other entities has to be avoided to reduce the risk of damage. In this paper, we approach this problem by applying deep reinforcement learning to generate pushing actions for a robotic manipulator acting on a planar surface where objects have to be pushed to goal locations while avoiding other items in the same workspace. With the latent space learned from a depth image of the scene and other observations of the environment, such as contact information between the end effector and the object as well as distance to the goal, our framework is able to learn contact-rich pushing actions that avoid collisions with other objects. As the experimental results with a six degrees of freedom robotic arm show, our system is able to successfully push objects from start to end positions while avoiding nearby objects. Furthermore, we evaluate our learned policy in comparison to a state-of-the-art pushing controller for mobile robots and show that our agent performs better in terms of success rate, collisions with other objects, and continuous object contact in various scenarios.

## I. INTRODUCTION

Pushing is often used for re-positioning and re-orientating objects since it simplifies the object manipulation in comparison to pick-and-place approaches. Furthermore, pushing allows for moving large, heavy, and irregularly shaped, as well as small and fragile objects to target positions and can be used for reducing uncertainty in the position of objects [1]. Hereby, the term pushing is separated in non-prehensile pushing [2] and prehensile pushing (push-grasp) [3], [4]. For example, in limited space [5], [6] and when dealing with fragile objects, non-prehensile pushing is the preferred manipulation action, since grasping increases the risk of damage. In the past, pushing has been used to separate objects for better grasping [7], [8] or to sort objects from a table into a bin [9] and is assumed to be more time-efficient than grasping to overcome short distances [10].

In general, pushing actions should be contact-rich with smooth arm motions. Furthermore, the contact to other objects in the workspace should be avoided to prevent any damages and changes the configuration of the scene. While for a long time, pushing behaviors were created using expert knowledge in an analytical way, more and more work is focusing on reinforcement learning (RL) to solve this task. Especially the ability to learn from environment interactions

All authors are with the Humanoid Robots Lab, University of Bonn, Germany. This work has partially been funded by the European Commission under grant agreement number 964854 –RePAIR – H2020-FETOPEN-2018-2020 and by the Deutsche Forschungsgemeinschaft (DFG, German Research Foundation) under Germany's Excellence Strategy, EXC-2070 – 390732324 – Phenorob.

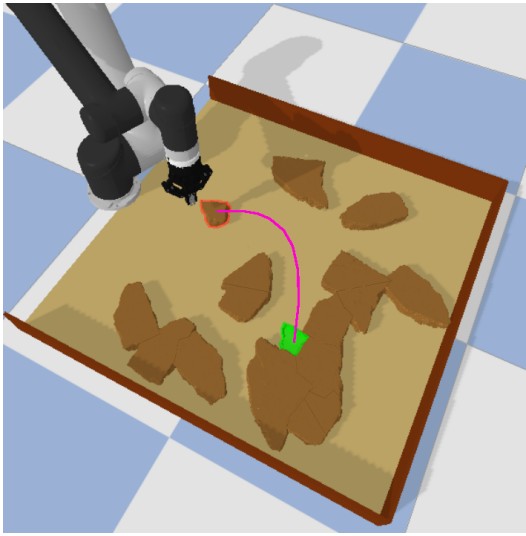

Fig. 1: Targeted application scenario of our system within the RePAIR-project[1]. The goal is to push the small fragment to the desired goal pose (green). Shown in magenta is the best pushing path, which maintains a safety distance to the other objects.

and own experiences makes RL a useful way to learn challenging new skills. Start-to-goal pushing with a RL-agent has been tackled before [13] and serves as a benchmark for RL [14], however, pushing in cluttered environments where collisions with other objects have to be avoided is a less researched area. While there are already approaches for mobile bases [15], [16], they have not been transferred to robotic manipulators so far.

In this paper, we present a framework to train a RL-agent that is able to realize obstacle-aware pushing in a contact-rich manner to guide objects with initially unknown dynamics on a planar surface to desired target configurations with a robotic manipulator. As representation of the workspace, we use a depth image taken from a bird's eye view. To reduce the size of the observation space and therefore the complexity, we use the latent space of a variational autoencoder. To accelerate learning, we calculate subgoals from an optimal 2D path in a grid representation of the environment generated from the depth image. In addition, we use further observations, such as contact information between the end effector of the manipulator and the object as well as the distance to the goal. The output of our system is an incremental motion of the current $(x, y, \theta)$-position of the robot's end effector. Fig. 1 illustrates a targeted application scenario from

the RePAIR-project[1]. The goal is to push the small fresco fragment to the desired position in a gentle manner while not damaging it or any other fragment on the assembly table.

Regarding related work, we differ compared to Bejjani *et al.* [4] in terms of object avoidance, use relative coordinates, like Migimatsu *et al.* [17] and include tactile force information, as proposed by Lee *et al.* [18] and Lin *et al.* [13]. Furthermore, we used for our approach ideas from Krivic *et al.* [15] and Regier *et al.* [19].

The key contributions of our work are the following

- A model-free RL system that learns to generate smooth pushing paths, with contact-rich pushing actions to reach the object's target positions in cluttered environments, thereby avoiding contact to other, nearby objects.
- A qualitative and quantitative evaluation in simulation in comparison to a state-of-the-art pushing controller [15], which we adapted to our scenario.

As the experiments show, our system leads to reliable pushing, while achieving better performance compared to [15] with respect to success rate, collisions with other objects, and continuous object contact in various scenarios.

## II. PROBLEM DESCRIPTION

In this work we consider the following problem. In a tabletop environment, a robotic arm is supposed to move an object from its current position to a 2D goal configuration. To achieve this, we consider the end effector (EE) of the arm moving in planar space $(x, y, \theta)$. The robotic arm can be of any degree of freedom (DOF). In addition to the pushing object, there are other objects which need to be considered as obstacles and which might obstruct the direct path to the end configuration. The obstacles have to be avoided by the EE and the object at all time. The goal of the RL-agent is to determine the best incremental movement $(\Delta x, \Delta y, \Delta \theta)$ of its EE position at each time step, to move the object with the EE as fast, but also as safe as possible to the goal position while avoiding obstacles on the way. An RGB-D camera is mounted centered above the scene in bird's eye view to obtain observations of the objects in the workspace.

## III. OUR APPROACH

We apply deep reinforcement learning to solve the task described above. This is motivated by the fact that we expect to obtain smoother trajectories as we would get with a pure control-based approach. Especially for traversing narrow passages, the lack of parameter tuning can be beneficial. We use a variational auto encoder (VAE) to decouple the feature extraction of the given depth image from the policy learning process [20]. Fig. 2 shows an overview of our proposed system. In the following, we describe our RL framework in detail.

[1]https://www.repairproject.eu

### A. Reinforcement Learning

For the implementation, we followed some ideas proposed by Regier *et al.* [19], which proposed a RL-framework to navigate in cluttered environments with a mobile robot. In the following we define the action and observation space, the reward function, the used RL-algorithm, the experience replay buffer strategy, as well as the learning strategy.

*1) Action Space:* Our action space consists of the three values, $(\Delta x, \Delta y, \Delta \theta)$, which are the increment to the current $x$ and $y$ position, as well as the yaw angle $\theta$ of the gripper. We set the maximum value of $(\Delta x, \Delta y, \Delta \theta)$ to the maximum distance change possible during one predefined time window.

*2) Observation Space:* The observation space of our RL-agent consists of 49 values, namely: The latent space (32), the EE position (5), the 6D joint angles (6), the 2D subgoal at t-1 and t-5 ($2 * 2$), an EE contact with obstacle indication (1) and the object to goal distance (1). To give the agent an indication of the best pushing direction, we include two subgoals of different time steps into the observation that we calculate from the current shortest path. The shortest path is calculated on a binary map, gathered from the depth image, where all obstacles are inflated according to half of the object's diameter. The agent never receives the complete shortest path in its observation and re-calculates it at each time step. Therefore, our agent is not constructed as a path following agent but learns the best pushing behavior during training.

*3) Reward:* Our reward function consists of following three components:

$$r_{dist} = \begin{cases} 50, & \text{if goal reached} \\ -r_{g\_dist} - r_{o\_dist}, & \text{otherwise} \end{cases} \quad (1)$$

$$r_{collison} = \begin{cases} -10, & \text{if object out of bounds} \\ -5 & \text{if collision occurred} \end{cases} \quad (2)$$

$$r_{touch} = \begin{cases} r_{o\_dist}, & \text{contact to object} \\ 0 & \text{otherwise} \end{cases} \quad (3)$$

The first equation encourages the agent toward a faster learning behavior since it penalizes higher distances between object and goal as well as object and EE with $r_{g\_dist}$ and $r_{o\_dist}$. $r_{collison}$ penalizes each collision of the object with clutter in the scene or if the object gets pushed out of bounds. The last part of the reward $r_{touch}$ considers the suggestion of Lin *et al.* [13]. Since we calculate the distance between the EE and the center of the object, a small distance value remains, even if the EE has contact to the object. Therefore, we negate the $r_{o\_dist}$ penalty of $r_{dist}$ each time the EE has contact to the object, to encourage a contact-rich behavior.

Together all three parts form the reward function $r_{total}$ of our agent:

$$r_{total} = r_{dist} + r_{collision} + r_{touch}$$

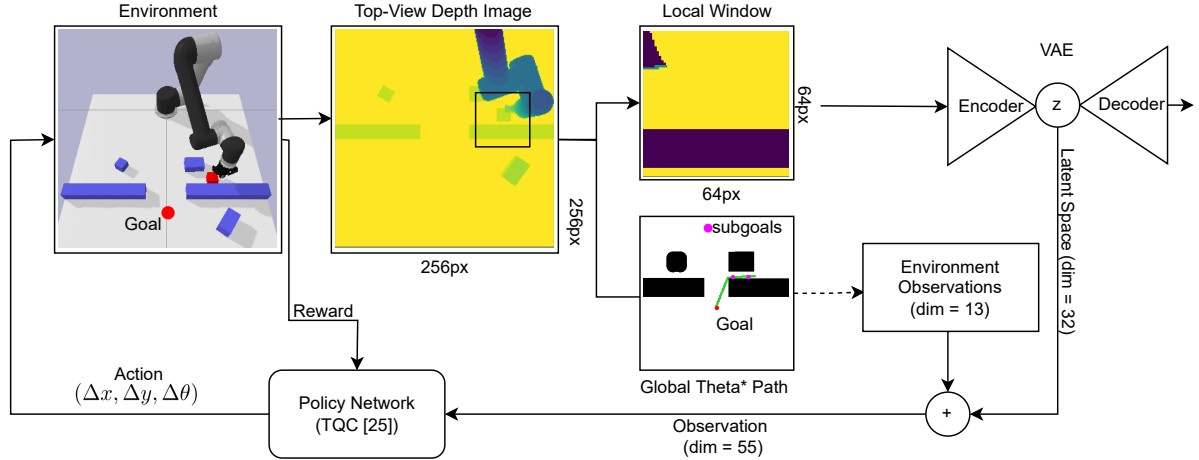

Fig. 2: Overview of our deep reinforcement learning pushing framework. Our system receives a depth image of the environment taken from an RGB-D camera. We calculate an object centered egocentric local window and feed it into the variational auto encoder to get the latent space. Furthermore, a global path from the current object position to the goal position, including subgoals, is calculated. The latent space, the subgoals, and further observations from the environments are used as the concatenated observation for the policy network of the deep RL which calculates the best 3D incremental motion.

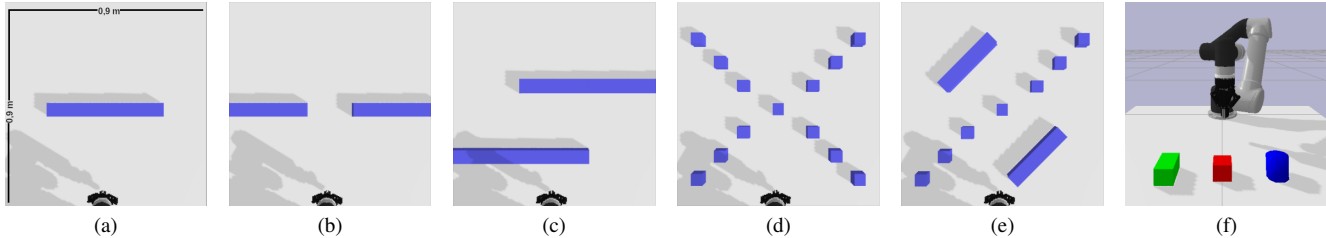

Fig. 3: Figures (a) to (c) depict example environments used for training and the quantitative evaluation. (d) and (e) shows unseen, complex environments to further evaluate the performance of our system and (f) the objects to be pushed. All objects have the same weight but differ in their geometrical shape. As pushing object during training we used the red cube. In a curriculum learning manner, we rotated the obstacle in (a) and vary its size during the training. Furthermore, the distance between the obstacles in (b) and (c) decreased from 20 cm to 10 cm, making the task more difficult.

*4) RL-Algorithm:* We evaluated two popular off-policy algorithms, namely Soft Actor-Critic [21] and Twin Delayed Deep Deterministic policy gradient (TD3) [22], as well as Truncated Quantile Critics (TQC) [23]. During our experiment, TQC led to the best and the most reproducible results and is therefore used for our experiments.

*5) Attentive Experience Replay:* The experience replay strategy enables agents to learn from previous experiences they made while interacting with the environment. We use the Attentive Experience Replay (AER) strategy, which samples entries in the replay buffer according to the similarities between the entry's state and the current state of the agent.

*6) Learning Strategy:* As the agent's learning strategy, we chose curriculum learning, which divides the task into subtasks and learns the subtasks one after another in increasing difficulty. We began the training with a maximum start-goal Euclidean distance of 0.06 m and increase it during training to up to 0.6 m. As training environments, we used the scenes shown in Fig. 3. The agent was trained for $7e6$ iterations. Without curriculum learning, the agent was not able to learn the task.

## IV. EXPERIMENTS

The goal of our experiments is to demonstrate the performance of our system qualitatively and quantitatively in free space as well as in obstacle-laden environments in terms of success rate, object contact, number of collisions, and shortest path deviation, i.e., normalized inverse path length (SPL) [24]. Furthermore, we provide a comparative evaluation against a state-of-the-art pushing control approach by Krivic *et al.* [15]. We performed the evaluation in pybullet [25] with a 6 DOF UR5[2] with a Robotiq 2f85 two-finger gripper[3]. The implementation of our learning framework with all hyperparameters as well as the reimplementation of the baseline approach is available at GitHub[4].

### A. Quantitative Evaluation

The quantitative evaluation consists of three parts, i.e., pushing with known and unknown objects in scenes with obstacles, and in previously unseen, highly cluttered scenes. All metrics except the success rate and the SPL are evaluated only on episodes that both methods could solve successfully.

---

[2]https://www.universal-robots.com/products/ur5-robot/
[3]https://robotiq.com/products/2f85-140-adaptive-robot-gripper
[4]https://github.com/NilsDengler/cluttered-pushing

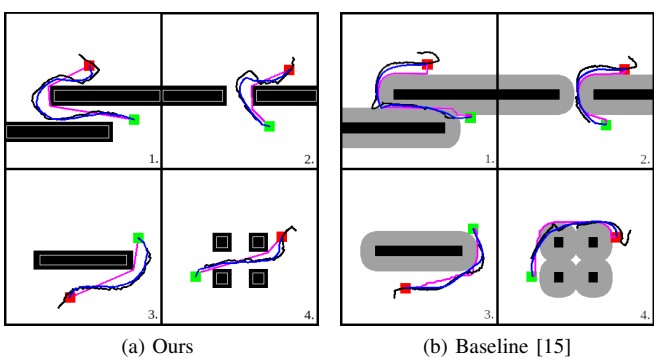

| (a) Ours | (b) Baseline [15] |

Fig. 4: Qualitative results from the quantitative evaluation of our approach (a) in comparison to the baseline [15] (b). Red indicates the start, green the goal position and magenta is the initial shortest path calculated by Lazy Theta* [26]. The path taken by the end effector is shown in black and the path of the object in blue. The grey area in (b) shows the increased traversal costs around obstacles, used for the baseline approach, while the obstacles in our approach (a) are inflated only by a small amount according to the half of the object's diameter.

| object avoidance | Success Rate | Object Contact Rate ∗ | Collision Rate ∗ | SPL | Path Length ∗ |
|---|---|---|---|---|---|
| Ours | **0.980** | **0.995** ± 0.02 | **0.008** ± 0.04 | 0.910 | 0.523 ± 0.18 |
| Krivic *et al.* [15] | 0.955 | 0.850 ± 0.10 | 0.011 ± 0.05 | **0.952** | **0.513**± 0.16 |

(a)

| Fragment | Success Rate | Object Contact Rate ∗ | Collision Rate ∗ | SPL | Path Length ∗ |
|---|---|---|---|---|---|
| Ours | 0.867 | **0.980** ± 0.05 | 0.05 ± 0.11 | 0.71 | 0.630± 0.29 |
| Ours re-trained | 0.867 | **0.980** ± 0.05 | 0.05 ± 0.11 | 0.71 | 0.630± 0.29 |
| Krivic *et al.* [15] | **0.953** | 0.868± 0.11 | **0.024**± 0.07 | **0.951** | **0.501** ± 0.16 |

(b)

| complex task | Success Rate | Object Contact Rate ∗ | Collision Rate ∗ | SPL | Path Length |
|---|---|---|---|---|---|
| Ours | **0.88** | **0.977** ± 0.06 | 0.065 ± 0.13 | **0.779** | **0.492**± 0.13 |
| Krivic *et al.* [15] | 0.72 | 0.566 ± 0.11 | **0.01** ± 0.05 | 0.720 | 0.550 ± 0.18 |

(c)

TABLE I: Quantitative evaluation of pushing in cluttered environments,. wrt. success rate, object contact, collisions, normalized inverse path length (SPL), and path length in meters. The values are the average over 500 runs for (a) and (b) and 50 for (c). The results are in comparison to the approach by Krivic *et al.* [15] where the metrics marked with a ∗ are significant according to the paired t-test with a chosen p-value of 0.05. As shown, our approach achieves overall better results in terms of success rate, object contact rate, and collision rate. Please refer to the text for more details.

The object contact rate is evaluated for each episode, once the EE first touched the object. Both, object contact rate and collision rate are the average of each episode, averaged over all episodes. For all experiments, we randomly sampled the distance between start and goal within 0.2 to 0.6 m. As pushing object during training, we used the red object shown in Fig. 3.

*1) Pushing in Scenes With Obstacles:* We generated different environments as shown in Fig 3 (a) to (c) and evaluated them with the learned object shown in (f), together with the completely unknown complex fragment shown in Fig. 1. We sampled the orientation and size of the obstacle in (a) as well as the distance between the obstacles in (b) to (c). For each object, we randomly generated 1,000 start-goal

configurations within the randomly sampled environments. The results are shown in Tab. Ia and b. With both objects, our approach achieves a significantly higher object contact rate in comparison to the baseline, which shows the benefit of our approach in terms of gentle pushing through contact-rich behavior. In terms of the SPL, the baseline achieves better results while there is no significantly increased path length. The higher SPL can be explained with the higher obstacle inflation necessary for the baseline approach and is illustrated in Fig. 4 that depicts example trajectories of the experiments. As can be seen, our agent has learned to safely navigate around objects, without strictly following the initial shortest path. This is a key advantage in comparison to the baseline approach, which follows the shortest path as tight as possible due to the properties of the controller method and is crucial if the parameters are not fine-tuned. Example 4 of Fig. 4 shows a scenario where our agent pushes a more efficient path, since it does not rely on any cost map and therefore on no parameter tuning. As the fragment was never seen during training, we retrained the agent and achieved similar results as with the red object. This underlines, that our system can be used for serving a general purpose but also retrained to specify on given scenarios.

*2) Pushing in Unseen, Complex Environments :* Finally, we designed more complex tasks with the goal to evaluate the capabilities of our trained agent in unseen environments with a higher density of clutter. We randomly sampled 50 start-goal configurations of the two scenarios (Fig. 3d and e), which contain many narrow passages. The results in Tab. Ic show the good performance in complex and completely unseen environments. Our agent achieved better results than Krivic *et al.* [15] in each metric except the collision rate. Especially, the contact rate is significantly increased. As already mentioned, our agent has not been trained on such scenarios, accordingly, the success rate is a bit lower in comparison to the other evaluations with the small cube. Regier *et al.* [19] showed that the agent will benefit, if it continues training in the unknown environment for a short time period.

## V. CONCLUSION

We presented a novel deep reinforcement learning approach for object pushing in cluttered tabletop environments. We demonstrated the efficacy of our approach in multiple simulated experiments, where the results show the increased performance in comparison to an existing control-based method with respect to various metrics. We showed that the pushing behavior highly benefits from our learning approach in terms of constant object contact and smooth trajectories avoiding obstacles while maintaining equal path length in comparison to the baseline method [15]. The evaluation of the runtime highlights that our system is capable of online pushing. The code of our system can be found on Github[4] and a video on our web page[5].

[5]https://www.hrl.uni-bonn.de/publications/dengler22iros-final.mp4

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
