# OpenReview forum: "Learning Goal-Oriented Non-Prehensile Pushing in Cluttered Scenes"
_ICRA.org/2022/Workshop/Contact-Rich — ICRA 2022 Workshop: RL for Manipulation Poster_

### Official Review · Reviewer_uNmw · 2022-05-09
**Novel and effective framework for contact-rich pushing in cluttered environments**

**Rating:** 9
**Confidence:** 3

**Review:**

Summary: This work proposed to use DRL for contact-rich pushing in cluttered environments. The policy takes local observations of robot, environment, contact, and global subgoals to maximize the reward to achieve the goal collision-free, and encourages contact during pushing. The framework is optimized with techniques like VAE, and curriculum learning. The experiments show the effectiveness of the framework compared to a state-of-the-art method.

Pros:
- This paper is well-written and pleasant to read.
- The proposed method nicely hybrids the shortest path for global planning, and RL for handling collision-free contact-rich local pushing primitives.
- The proposed method effectively encourages contact-rich behavior.
- The experiments show the proposed methods achieve better performance than the baseline in several aspects.

Cons:
- The work can be strengthened with experiments/discussions about the real-robot scenarios, including the sim2real gap, data efficiency, limitation, etc.

---

### Official Review · Reviewer_vWPZ · 2022-05-10
**Learning Goal-Oriented Non-Prehensile Pushing in Cluttered Scenes - review**

**Rating:** 8
**Confidence:** 4

**Review:**

This paper uses Deep RL to train a robot with end-effector control in a cluttered pushing task where the object is moved to a goal object position and orientation with minimal contact with other parts of the environment. Their experiments compare a trained RL policy and a pushing baseline from Krivic and Piater '19. They train the policy using a reward function for contact-rich interaction without collisions with the environment, and a curriculum which gradually increases the distance between the starting point and the goal. As input, they concatenate a 32-dim latent space from a VAE encoding of RGB-D observations from an overhead camera and ground-truth environment observations that include subgoal information to reach the goal. In the experiments, they test the system with variations of the original training environment, including more complex obstacle geometries.

One claim made in their results states that their method can be trained to solve the task generally without parameter tuning, and also be retrained to improve performance on specific scenarios. However, this claim is not clearly supported in Table 1(b), since the results for their method before and after re-training are identical. It would be helpful if this claim is updated, and if the subtables in Table 1 each included their own caption to more clearly indicate which experiments/environments they correspond to. Otherwise, the experimental and system design are clearly explained, and their method shows a improvement over the baseline in the unseen environment case. It might be worth showing whether re-training/finetuning to the new environment shows an improvement, and what the collision rate is during re-training, to highlight another clear advantage their method has to the baseline.